# Survey report on keratoplasty in China: A 5-year review from 2014 to 2018

Hua Gao[1,2], Ting Huang[2,3], Zhiqiang Pan[2,4], Jie Wu[2,5], Jianjiang Xu[2,6], Jing Hong[2,7], Wei Chen[2,8], Huping Wu[9], Qian Kang[10], Lei Zhu[2,11], Lingling Fu[12], Liqiang Wang[2,13], Guigang Li[2,14], Zhihong Deng[15], Hong Zhang[2,16], Hui Xu[17], Qingliang Zhao[18], Hongshan Liu[19], Linnong Wang[2,20], Baihua Chen[2,21], Xiuming Jin[2,22], Minghai Huang[23], Jizhong Yang[24], Minghong Gao[2,25], Wentian Zhou[26], Hanping Xie[2,27], Yao Fu[2,28], Feng Wen[29], Changbo Fu[30], Shaozhen Zhao[2,31], Yanning Yang[2,32], Yanjiang Fu[33], Tao Yao[34], Chaoqing Wang[35], Xiaonan Sun[36], Xiaowei Gao[2,37], Maimaitiming Reziwan[38], Yingping Deng[2,39], Jian Li[40], Limei Liu[41], Bo Zeng[42], Lianyun Bao[43], Hua Wang[2,44], Lijun Zhang[2,45], Zhiyuan Li[46], Zhijian Yin[47], Yuechun Wen[48], Xiao Zheng[49], Liqun Du[50], Zhenping Huang[51], Xunlun Sheng[52], Hui Zhang[2,53], Lizhong Chen[54], Xiaoming Yan[2,55], Xiaowei Liu[56], Wenhui Liu[57], Yuan Liu[58], Liang Liang[59], Pengcheng Wu[60], Lijun Qu[61], Jinkui Cheng[62], Hua Zhang[63], Qige Qi[64], Yangkyi Tseten[65], Jianping Ji[3], Jin Yuan[2,3], Ying Jie[4], Jun Xiang[6], Yifei Huang[2,13], Yuli Yang[2,27], Ying Li[2,56], Yiyi Hou[1], Tong Liu[1], Lixin Xie[2,66]*, Weiyun Shi[1,2]*

**Data Availability Statement:** All relevant data are within the manuscript and its Supporting Information files.

**Funding:** Funding: This study was supported by National Natural Science Foundation of China,

**1** Department of Ophthalmology, Eye Hospital of Shandong First Medical University, State Key Laboratory Cultivation Base, Shandong Provincial Key Laboratory of Ophthalmology, Shandong Eye Institute, Shandong First Medical University & Shandong Academy of Medical Sciences, Jinan, China, **2** Corneal Disease Group of Ophthalmological Society of Chinese Medical Association (China Cornea Society), Jinan, China, **3** Department of Ophthalmology, State Key Laboratory of Ophthalmology, Zhongshan Ophthalmic Centre, Sun Yat-sen University, Guangzhou, China, **4** Department of Ophthalmology, Beijing Tongren Hospital, Capital Medical University, Beijing, China, **5** Department of Ophthalmology, No.1 Hospital of Xi'an City, Xian, China, **6** Department of Ophthalmology, Eye and Ear, Nose, Throat Hospital, Fudan University, Shanghai, China, **7** Department of Ophthalmology, Peking University Third Hospital, Beijing, China, **8** Department of Ophthalmology, Eye Hospital of Wenzhou Medical University, Wenzhou, China, **9** Department of Ophthalmology, Affiliated Xiamen Eye Center of Xiamen University, Xiamen, China, **10** Department of Ophthalmology, Chengdu AiDi Eye Hospital, Chengdu, China, **11** Department of Ophthalmology, Henan Eye Institute, Henan Eye Hospital, Zhengzhou, China, **12** Department of Ophthalmology, Hefei Puri Ophthalmological Hospital, Hefei, China, **13** Department of Ophthalmology, Chinese PLA General Hospital, Medical School of Chinese PLA, Beijing, China, **14** Department of Ophthalmology, TongJi Hospital, Tongji Medical College, Huazhong University of Science and Technology, Wuhan, China, **15** Department of Ophthalmology, The Third Xiangya Hospital of Central South University, Changsha, China, **16** Department of Ophthalmology, The First Affiliated Hospital of Harbin Medical University, Harbin, China, **17** Department of Ophthalmology, The First Hospital of Jilin University, Changchun, China, **18** Department of Ophthalmology, Suzhou Lixiang Eye Hospital, Suzhou, China, **19** Department of Ophthalmology, Hainan Eye Institute, Hainan Eye Hospital, Haikou, China, **20** Department of Ophthalmology, Nanjing First Hospital, Nanjing Medical University, Nanjing, China, **21** Department of Ophthalmology, Second Xiangya Hospital, Central South University, Changsha, China, **22** Department of Ophthalmology, Affiliated Second Hospital, School of Medicine, Zhejiang University, Hangzhou, China, **23** Department of Ophthalmology, Nanning Aier Eye Hospital, Nanning, China, **24** Department of Ophthalmology, Shanxi Eye Hospital, Taiyuan, China, **25** Department of Ophthalmology, General Hospital of Northern Theater Command Hospital, Shenyang, China, **26** Department of Ophthalmology, Affiliated Eye Hospital of Nanchang University, Nanchang, China, **27** Department of Ophthalmology, The First Hospital Affiliated to Army Medical University, Chongqing, China, **28** Department of Ophthalmology, Shanghai Ninth People's Hospital, Shanghai JiaoTong University School of Medicine, Shanghai, China, **29** Department of Ophthalmology, Ningbo Eye Hospital, Ningbo, China, **30** Department of Ophthalmology, Xuzhou Municipal Hospital Affiliated to Xuzhou Medical University, Xuzhou, China, **31** Department of Ophthalmology, Tianjin Medical University Eye Hospital, Tianjin, China, **32** Department of Ophthalmology, Renmin Hospital of Wuhan University, Wuhan, China, **33** Department of Ophthalmology, Daqing Ophthalmologic Hospital, Daqing, China, **34** Department of Ophthalmology, Shenyang He Eye Hospital, Shenyang, China, **35** Department of Ophthalmology, Jinan Mingshui Eye Hospital, Jinan, China, **36** Department of Ophthalmology, The 4th People's Hospital of Shenyang, Shenyang, China, **37** Department of Ophthalmology, 474 Hospital of PLA, Urumchi, China, **38** Department of Ophthalmology, Urumqi City Ophthalmology and Otolaryngology Hospital, Urumchi, China, **39** Department of

grant number: 81570821 (HG), 81870639(HG); Key Project of National Natural Science Foundation of China, grant number: 81530027 (WS); Taishan Scholar Program, grant number: 20150215 (WS), 201812150(HG); and the Innovation Project of Shandong Academy of Medical Sciences (HG). The funders had no role in study design, data collection and analysis, decision to publish, or preparation of the manuscript.

**Competing interests:** Competing interests: The authors have declared that no competing interests exist.

Ophthalmology, West China Hospital of Sichuan University, Chengdu, China, **40** Department of Ophthalmology, The First Affiliated Hospital of Nanjing Medical University, Jiangsu Province Hospital, Nanjing, China, **41** Department of Ophthalmology, Weifang Eye Hospital, Weifang, China, **42** Department of Ophthalmology, General Hospital of the Central Theater of the Chinese People's Liberation Army, Wuhan, China, **43** Department of Ophthalmology, Nanjing Ningyi Eye Center, Nanjing, China, **44** Department of Ophthalmology, Xiangya Hospital, Central South University, Changsha, China, **45** Department of Ophthalmology, The People's Third Hospital of Dalian, Dalian Medical University, Dalian, China, **46** Department of Ophthalmology, The People's No.1 Hospital of Chenzhou, Chenzhou, China, **47** Department of Ophthalmology, The First Affiliated Hospital of Dali University, Dali, China, **48** Department of Ophthalmology, The First Affiliated Hospital of University of Science and Technology of China, Anhui Provincial Hospital, Hefei, China, **49** Department of Ophthalmology, The Army Characteristic Medical Center, Chongqing, China, **50** Department of Ophthalmology, Qilu Hospital of Shandong University, Jinan, China, **51** Department of Ophthalmology, Jinling Hospital, Nanjing University School of Medicine, Nanjing, China, **52** Department of Ophthalmology, Ningxia Eye Hospital, The People's Hospital of Ningxia Hui Autonomous Region, Yinchuan, China, **53** Department of Ophthalmology, First Affiliated Hospital of Kunming Medical University, Kunming, China, **54** Department of Ophthalmology, Lunan Eye Hospital, Linyi, China, **55** Department of Ophthalmology, Peking University First Hospital, Beijing, China, **56** Department of Ophthalmology, Peking Union Medical College Hospital, Chinese Academy of Medical Sciences and Peking Union Medical College, Beijing, China, **57** Department of Ophthalmology, Wuxi Second People's Hospital, Wuxi, China, **58** Department of Ophthalmology, Guizhou Jinglang Eye Hospital, Guiyang, China, **59** Department of Ophthalmology, Yichang Central People's Hospital, Yichang, China, **60** Department of Ophthalmology, Lanzhou University Second Hospital, Lanzhou, China, **61** Department of Ophthalmology, The Second Affiliated Hospital of Harbin Medical University, Harbin, China, **62** Department of Ophthalmology, Jingzhou First People's Hospital, Jingzhou, China, **63** Department of Ophthalmology, Shijiazhuang No.1 Hospital, Shijiazhuang, China, **64** Department of Ophthalmology, Hulunbuir People's Hospital, Hulunbuir, China, **65** Department of Ophthalmology, Tibetan Traditional Tibet Medical Hospital of Tibet Autonomous Region, Lhasa, China, **66** Department of Ophthalmology, Qingdao Eye Hospital of Shandong First Medical University, State Key Laboratory Cultivation Base, Shandong Provincial Key Laboratory of Ophthalmology, Shandong Eye Institute, Shandong First Medical University & Shandong Academy of Medical Sciences, Qingdao, China

☯ These authors contributed equally to this work.
* weiyunshi@163.com, wyshi@sdfmu.edu.cn(WS); lixinxie@public.qd.sd.cn(LX)

## Abstract

To provide the general information on corneal transplantation (CT) in China, China Cornea Society designed a questionnaire on CT from 2014 to 2018 and entrusted it to 31 committee members for implementation of the survey nationwide. This article presents the results of the survey and compares the indicators used in the survey and those in the annual statistical report released by the Eye Bank Association of America (EBAA). The number of corneal transplantations completed by the 64 hospitals from 2014 to 2018 was respectively 5377, 6394, 7595, 8270 and 8980, totally 36,616 (22,959 male and 13,657 female). The five largest hospitals by the number of corneal transplantations completed 15,994 surgeries in total, accounting for 43.68% of all the surgeries performed in the 64 hospitals. The most common indication for corneal transplantations was corneal leukoma (7683, 20.98%), followed by bacterial keratitis (4209, 11.49%), corneal dystrophies (4189, 11.44%), keratoconus (3578, 9.77%) and corneal perforation (2839, 7.75%). The main surgical techniques were penetrating keratoplasty (PK) (19,896, 54.34%), anterior lamellar keratoplasty (ALK) (13,869, 37.88%). The proportion of PK decreased from 57.97% in 2014 to 52.88% in 2018 while the proportion of ALK increased from 36.04% in 2014 to 37.92% in 2018. The geographical distribution of keratoplasties performed in China is unbalanced. PK and ALK were the main techniques of CT and corneal leukoma, bacterial keratitis and corneal dystrophies were the main indications for CT in China.

## Introduction

According to the World Health Organization, corneal diseases are one of the leading causes of blindness globally [1]. Approximately 180,000 cases of corneal transplantations are performed worldwide each year [2], of which 40,000 to 50,000 cases are in the United States, restoring vision for patients with corneal blindness [3,4]. China is the largest and most populous developing country in the world and corneal diseases are the second leading causes of blindness [5–7]. According to the multi-center study on infectious keratitis in China conducted by Song et al. in 2010, the number of people with corneal blindness in at least one eye in China was estimated to be about 3 million at that time [8].

Eye Bank Association of America (EBAA) is the only accredited organization in America that distributes donated eye tissues and collects information on the utilization of the tissues in all its member banks. Two articles based on the statistical reports of EBAA has been published, reporting the trends of penetrating keratoplasty (PK) from 1980 to 2004 and the condition of keratoplasty in the United States from 2005 through 2014 respectively [3,4]. All these data can provide information for ophthalmologists to understand the general situation of corneal transplantation (CT), and play a positive role in promoting the development of CT and the improvement in surgical techniques.

Although several single center based articles about keratoplasties have been published these years [9–14], currently, there is no national eye bank association or eye bank union in China, therefore, no exact data is available for the total number of corneal transplantations each year, the main indications for CT, surgical techniques of CT, etcetera, which limits the formulation and implementation of corneal blindness-related policies and then the process of restoring vision for the blind in China. To get an overview of keratoplasties in China, the Corneal Disease Group of Ophthalmological Society of Chinese Medical Association (China Cornea Society) designed a questionnaire on CT (S1 Table) and entrusted the questionnaire to its 31 committee members for the implementation of the survey in hospitals. 64 hospitals participated in the survey and returned the questionnaire. The results of the survey show that there are many differences in surgical techniques of CT, indications, and other aspects between China and America. The survey data are now available for reference.

## Materials and methods

The China Cornea Society entrusted the questionnaire to its 31 committee members for the implementation of the survey in China. A total number of 64 hospitals were involved in the survey and asked to fill in the questionnaires. This study was approved by the Institutional Review Board of Shandong Eye Institute and adhered to the tenets of the Declaration of Helsinki.

Indicators of the survey included the number of keratoplasties performed between January 2014 to December 2018, general conditions of each patient, geographical distribution of surgeries, indications for surgeries, surgical techniques. Detailed data of each case was collected.

### Indications for corneal transplantation

Indications for CT mainly include 1) fungal keratitis, 2) bacterial keratitis, 3) herpes simplex virus keratitis, 4) Acanthamoeba keratitis), 5) corneal dystrophies (granular dystrophy, lattice dystrophy, macular dystrophy, Fuchs' dystrophy, and other dystrophies), 6) trauma (acid burns, alkali burns, thermal burns, and other burns), 7) immune-related keratitis (Mooren's Ulcer, rheumatoid arthritis-associated peripheral ulcerative keratitis, and Stevens–Johnson syndrome), 8) corneal degeneration, 9) keratoconus, 10) pseudophakic bullous keratopathy (PBK), 11) corneal tumor, 12) corneal leukoma, 13) corneal perforation, 14) corneal staphyloma, 15) exposure keratitis, 16) pterygium-related corneal opacity, and 17) graft opacity.

### Corneal transplantation techniques

Corneal transplantation techniques mainly include 1) PK, 2) lamellar keratoplasty (LK), in the form of anterior lamellar keratoplasty (ALK) and endothelial keratoplasty (EK), 3) kerato-prosthesis, and 4) keratolimbal allograft.

### Statistical method

The data were analyzed using SPSS Statistics version 25 (IBM, Armonk, NY). P values less than 0.05 were regarded as statistically significant.

## Results

64 tertiary hospitals in China returned the questionnaires. According to the detailed information the hospitals provided, the number of CT performed in the five years was 5377, 6394, 7595, 8270, and 8980 respectively, totally 36,616.

### Number of corneal transplantations in hospitals and geographical distribution

The five largest hospitals by number of corneal transplantations performed from 2014 through 2018 are Shandong Eye Institute (including Eye Hospital of Shandong First Medical University and Qingdao Eye Hospital of Shandong First Medical Univeristy) (4001), Zhongshan Ophthalmic Center (3837), Beijing Tongren Hospital (3079), No. 1 Hospital of Xi'an City (2569), Eye and Ear, Nose, Throat Hospital of Fudan University (2508) and totally conducted 15,994 keratoplasties, accounting for 43.68% of the totally reported keratoplasties (Fig 1).

According to the location of the hospitals reported, 30 provincial-level administrative units were included. The total number of CT performed in the five years in each administrative unit is as follows (Table 1).

The five largest provincial-level administrative units by number of CT have completed 19 911 cases (54.38%), while the five smallest units have only completed 67 cases (0.18%).

### Age and gender distribution of cornea transplant recipients

Among the 36,616 patients who underwent CT, 22,959 (62.70%) were male and 13,657 (37.30%) were female. The distribution of age is as follows (Table 2).

### Indications for corneal transplantation

The 5 leading indications for CT in China were corneal leukoma (7683, 20.98%), bacterial keratitis (4209, 11.49%), corneal dystrophies (4189, 11.44%), keratoconus (3578, 9.77%), and corneal perforation (2839, 7.75%) (Fig 2).

### Surgical techniques

According to the data we collected, the number of PK accounts for 54.34% of all the keratoplasties performed in the five years, ALK 37.88%, EK 6.87%, keratoprosthesis 0.72%, and keratolimbal allograft 0.19% (Fig 3).

The most common surgical technique was PK, although decreasing from 57.97% in 2014 to 52.88% in 2018. ALK increased from 36.04% to 37.92% in the five years (P<0.001) and EK increased from 5.52% to 7.75% (P<0.001). The indications for ALK are shown in Table 3 and the indications for PK are shown in Table 4.

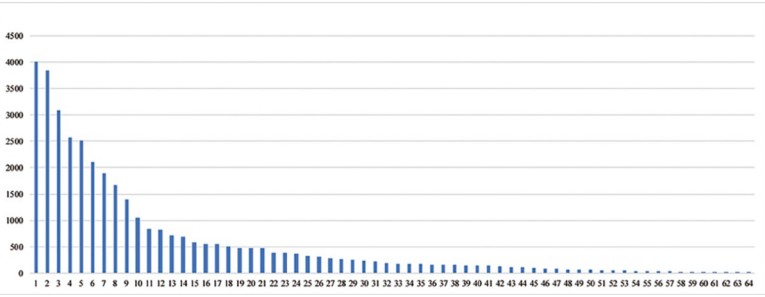

**Fig 1. Number of corneal transplants performed in each involved hospital.** 1: Shandong Eye Institute 2: Zhongshan Ophthalmic Centre, Sun Yat-sen University 3: Beijing Tongren Hospital, Capital Medical University 4: No.1 Hospital of Xi'an City 5: Eye and Ear, Nose, Throat Hospital, Fudan University 6: Peking University Third Hospital 7: Eye Hospital of Wenzhou Medical University 8: Affiliated Xiamen Eye Center of Xiamen University 9: Chengdu AiDi Eye Hospital 10: Henan Eye Institute, Henan Eye Hospital 11: Hefei Puri Ophthalmological Hospital 12: Chinese PLA General Hospital, Medical School of Chinese PLA 13: TongJi Hospital, Tongji Medical College, Huazhong University of Science and Technology 14: The Third Xiangya Hospital of Center South University 15: The First Affiliated Hospital of Harbin Medical University 16: The First Hospital of Jilin University 17: Suzhou Lixiang Eye Hospital 18: Henan Eye Institute, Henan Eye Hospital 19: Nanjing First Hospital, Nanjing Medical University 20: Department of Ophthalmology, Second Xiangya Hospital, Central South University 21: Affiliated Second Hospital, School of Medicine, Zhejiang University 22: Nanjing Aier Eye Hospital 23: Shanxi Eye Hospital 24: General Hospital of Northern Theater Command Hospital 25: Affiliated Eye Hospital of Nanchang University 26: The First Hospital Affiliated to Army Medical University 27: Shanghai Ninth People's Hospital, Shanghai JiaoTong University School of Medicine 28: Ningbo Eye Hospital 29: Xuzhou Municipal Hospital Affiliated to Xuzhou Medical University 30: Tianjin Medical University Eye Hospital 31: Renmin Hospital of Wuhan University 32: Daqing Ophthalmologic Hospital 33: Shenyang He Eye Hospital 34: Jinan Mingshui Eye Hospital 35: The 4th People's Hospital of Shenyang 36: 474 Hospital of PLA 37: Urumqi City Ophthalmology and Otolaryngology Hospital 38: West China Hospital of Sichuan University 39: The First Affiliated Hospital of Nanjing Medical University, Jiangsu Province Hospital 40: Weifang Eye Hospital 41: General Hospital of the Central Theater of the Chinese People's Liberation Army, Wuhan, Hubei Province 42: Nanjing Ningyi Eye Center 43: Department of Ophthalmology, Xiangya Hospital, Central South University 44: The People's Third Hospital of Dalian, Dalian Medical University, The third People's Hospital of Dalian 45: The People's No.1 Hospital of Chenzhou 46: The First Affiliated Hospital of Dali University 47: The First Affiliated Hospital of University of Science and Technology of China, Anhui Provincial Hospital 48: The Army Characteristic Medical Center 49: Qilu Hospital of Shandong University 50: Jinling Hospital, Nanjing University School of Medicine 51: Ningxia Eye Hospital, The People's Hospital of Ningxia Hui Autonomous Region 52: First Affiliated Hospital of Kunming Medical University 53: Lunan Eye Hospital 54: Peking University First Hospital 55: Peking Union Medical College Hospital, Chinese Academy of Medical Sciences and Peking Union Medical College 56: Wuxi Second People's Hospital 57: Guizhou Jinglang Eye Hospital 58: Yichang Central People's Hospital 59: Lanzhou University Second Hospital 60: The Second Affiliated Hospital of Harbin Medical University 61: Jingzhou First People's Hospital 62: Shijiazhuang No.1 Hospital 63: Hulunbuir People's Hospital 64: Tibetan Traditional Tibet Medical Hospital of Tibet Autonomous Region.

## Discussion

Corneal diseases are a leading cause of blindness worldwide, second to cataract [1,15]. Due to the unbalanced global economy and ethnic differences, the causes of blindness, the proportion of the causes, the preferred surgical techniques of CT and the indications for surgery are not the same or even quite different in different countries and regions [16–18]. Currently, China does not have a national eye bank union, so it is not easy to collect exact information on CT from all the hospitals and eye banks. Therefore, the general data on CT in China is unavailable now. The Chinese Ophthalmological Society is the most authoritative academic association in ophthalmology in China and its Corneal Disease Group (China Cornea Society) is the most authoritative group in the cornea disease field in China, whose members are experts on cornea-related clinical work and academic research. The members are very familiar with the hospitals in their own administrative unit, so they were entrusted to collect information from the hospitals in their own administrative units. The survey has covered most of the qualified hospitals in performing keratoplasties in the recent five years and it is estimated that the cases

**Table 1. Total number of CT performed in the five years in each provincial-level administrative unit.**

|  |  | Number* | % |
|---|---|---|---|
| Provincial-level administrative unit | Beijing | 6077 | 16.60 |
|  | Shandong | 4427 | 12.09 |
|  | Guangdong | 3976 | 10.86 |
|  | Shanghai | 2791 | 7.62 |
|  | Zhejiang | 2 640 | 7.21 |
|  | Shaanxi | 2 569 | 7.02 |
|  | Fujian | 1 671 | 4.56 |
|  | Jiangsu | 1 650 | 4.51 |
|  | Sichuan | 1 540 | 4.21 |
|  | Hunan | 1 371 | 3.74 |
|  | Henan | 1 043 | 2.85 |
|  | Hubei | 985 | 2.69 |
|  | Anhui | 917 | 2.50 |
|  | Liaoning | 825 | 2.25% |
|  | Heilongjiang | 787 | 2.15 |
|  | Jilin | 554 | 1.51 |
|  | Hainan | 505 | 1.38 |
|  | Guangxi | 392 | 1.07 |
|  | Chongqing | 384 | 1.05 |
|  | Shanxi | 379 | 1.04 |
|  | Jiangxi | 325 | 0.89 |
|  | Xinjiang | 313 | 0.85 |
|  | Tianjin | 241 | 0.66 |
|  | Yunnan | 134 | 0.37 |
|  | Ningxia | 53 | 0.14 |
|  | Guizhou | 34 | 0.09 |
|  | Gansu | 15 | 0.04 |
|  | Hebei | 8 | 0.02 |
|  | Inner Mongolia | 8 | 0.02 |
|  | Tibet | 2 | 0.01 |

* in descending order.

included in the survey accounted for more than 90% of all the corneal transplantations, so the results can represent the condition of keratoplasties in China.

According to our survey, the four leading indications for CT in China from 2014 through 2018 were corneal leukoma, bacterial keratitis, corneal dystrophies, and keratoconus, while the four leading indications for CT in America from 2005 to 2014 were PBK, keratoconus, Fuchs' dystrophy, and repeat CT [4]. Analyses of the results show that there are both similarity and difference on the indications for CT between China and America.

**Table 2. Age distribution of cornea transplant recipients.**

|  | Age (year) | | | | | | | | | |
|---|---|---|---|---|---|---|---|---|---|---|
|  | <1 | 1–10 | 11–20 | 21–30 | 31–40 | 41–50 | 51–60 | 61–70 | 71–80 | >80 |
| Number (%) | 538 (1.47) | 3123 (8.53) | 2846 (7.77) | 3607 (9.85) | 3427 (9.36) | 6053 (16.53) | 7067 (19.30) | 6438 (17.58) | 2851 (7.79) | 666 (1.82) |

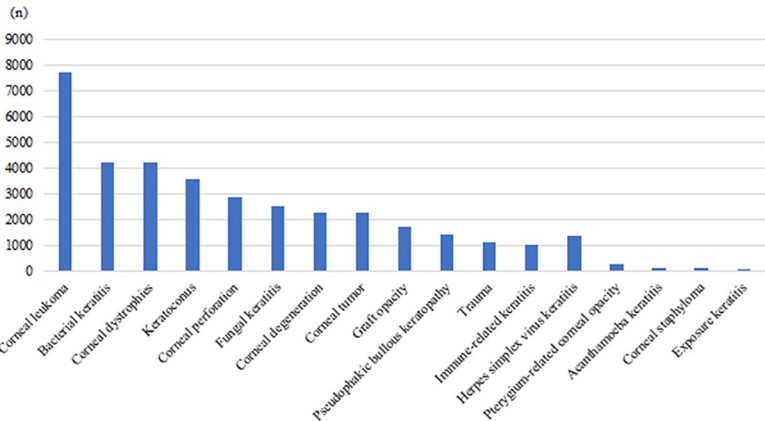

**Fig 2. Indications for corneal transplantation.**

Similarity: In both countries, keratoconus is among the top four indications of CT. Reason: Keratoconus occurs in all ethnic groups and it is related to heredity, therefore, the incidence rate of keratoconus in different countries is slightly different or comparable [19–22].

Difference: In America, endothelial dysfunction (including PBK and Fuchs' dystrophy) is the main indication for CT, while in China, corneal leukoma, and infectious keratitis are the most common indications. Reasons: 1) The difference may be related to the economic development levels of the two countries. China is a developing country with an agricultural population of 800 million. The economic level and geographical conditions limit the promotion and application of agricultural machinery and the people still lack occupational safety awareness

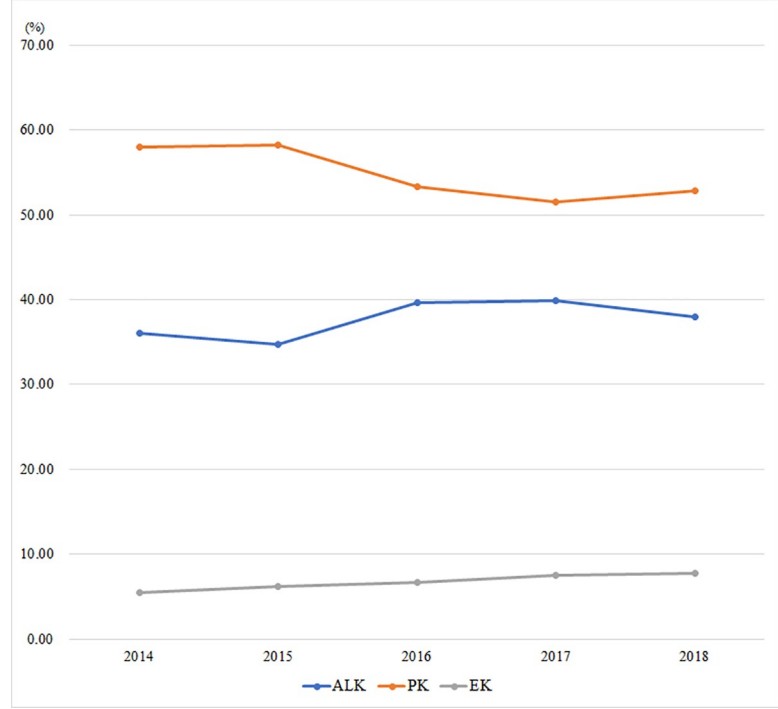

**Fig 3. Tendency of preferred surgical techniques.** ALK = anterior lamellar keratoplasty; PK = penetrating keratoplasty; EK = endothelial keratoplasty.

**Table 3. Indications for anterior lamellar keratoplasty from 2014 through 2018.**

| Indication | Year, Number (%) | | | | |
|---|---|---|---|---|---|
| | 2014 | 2015 | 2016 | 2017 | 2018 |
| Corneal leukoma | 280(14.45) | 338(15.23) | 387(12.86) | 591(17.93) | 455(13.36) |
| Keratoconus | 255(13.16) | 297(13.38) | 459(15.25) | 547(16.59) | 531(15.59) |
| Corneal tumor | 217(11.20) | 245(11.04) | 370(12.29) | 468(14.19) | 536(15.74) |
| Bacterial keratitis | 223(11.51) | 232(10.46) | 406(13.49) | 391(11.86) | 458(13.45) |
| Corneal degeneration | 164(8.46) | 151(6.80) | 245(8.14) | 323(9.80) | 296(8.69) |
| Corneal dystrophies | 88(4.54) | 158(7.12) | 211(7.01) | 154(4.67) | 283(8.31) |
| Corneal perforation | 126(6.50) | 144(6.49) | 199(6.61) | 176(5.34) | 167(4.90) |
| Immune-related keratitis | 103(5.31) | 161(7.26) | 185(6.15) | 121(3.67) | 141(4.14) |
| Fungal keratitis | 108(5.57) | 49(2.21) | 129(4.29) | 202(6.13) | 222(6.52) |
| Trauma | 107(5.52) | 129(5.81) | 157(5.22) | 104(3.15) | 98(2.88) |
| Herpes simplex virus keratitis | 94(4.85) | 131(5.90) | 122(4.05) | 77(2.34) | 92(2.70) |
| Graft opacity | 32(1.65) | 47(2.12) | 56(1.86) | 84(2.55) | 85(2.50) |
| Pseudophakic bullous keratopathy | 93(4.80) | 74(3.33) | 43(1.43) | 9(0.27) | 4(0.12) |
| Pterygium-related corneal opacity | 16(0.83) | 25(1.13) | 20(0.66) | 26(0.79) | 19(0.56) |
| Acanthamoeba keratitis | 23(1.19) | 27(1.22) | 10(0.33) | 11(0.33) | 10(0.29) |
| Corneal staphyloma | 7(0.36) | 8(0.36) | 8(0.27) | 13(0.39) | 8(0.23) |
| Exposure keratitis | 2(0.10) | 3(0.14) | 3(0.10) | 0(0) | 0(0) |

and protection measures. As a result, the risk of infectious keratitis caused by trauma in farming is high [23], therefore, corneal leukoma and infectious keratitis become the most common indications for CT in China. 2) America has a relatively long history in performing cataract surgeries and the total number of surgeries is large. The wide application of anterior chamber intraocular lenses in the 1980's has accelerated the loss of endothelial cells after surgery, resulting in a large amount of PBK, which was one of the most common indications for CT from

**Table 4. Indications for penetrating keratoplasty from 2014 through 2018.**

| Indication | Year, Number (%) | | | | |
|---|---|---|---|---|---|
| | 2014 | 2015 | 2016 | 2017 | 2018 |
| Corneal leukoma | 942(20.22) | 1043(28.04) | 1064(26.30) | 1238(29.03) | 1194(25.14) |
| Bacterial keratitis | 289(9.27) | 348(9.35) | 492(12.16) | 555(13.01) | 710(14.95) |
| Corneal perforation | 214(6.87) | 315(8.47) | 360(8.90) | 518(12.15) | 600(12.63) |
| Corneal dystrophies | 285(9.14) | 347(9.33) | 406(10.04) | 330(7.74) | 449(9.45) |
| Fungal keratitis | 257(8.25) | 196(5.27) | 382(9.44) | 456(10.69) | 466(9.81) |
| Keratoconus | 296(9.50) | 387(10.40) | 342(8.45) | 247(5.79) | 213(4.49) |
| Graft opacity | 208(6.67) | 250(6.72) | 271(6.70) | 285(6.68) | 313(6.59) |
| Corneal degeneration | 142(4.56) | 164(4.41) | 166(4.10) | 201(4.71) | 276(5.81) |
| Herpes simplex virus keratitis | 117(3.75) | 107(2.88) | 141(3.49) | 177(4.15) | 230(4.84) |
| Pseudophakic bullous keratopathy | 145(4.65) | 111(2.98) | 136(3.36) | 105(2.46) | 136(2.86) |
| Trauma | 71(2.28) | 100(2.69) | 101(2.50) | 85(1.99) | 85(1.79) |
| Corneal tumor | 92(2.95) | 129(3.47) | 78(1.93) | 30(0.71) | 33(0.69) |
| Immune-related keratitis | 39(1.25) | 65(1.75) | 65(1.68) | 24(0.56) | 28(0.59) |
| Pterygium-related corneal opacity | 1(0.03) | 144(3.87) | 8(0.20) | 0(0) | 2(0.04) |
| Corneal staphyloma | 9(0.29) | 13(0.35) | 18(0.44) | 7(0.16) | 8(0.17) |
| Acanthamoeba keratitis | 10(0.33) | 1(0.03) | 9(0.22) | 7(0.16) | 6(0.13) |
| Exposure keratitis | 0(0) | 0(0) | 3(0.07) | 0(0) | 0(0) |

2005 through 2014 [4]. Although the incidence rate of PBK has decreased worldwide these years with the development of facilities for cataract surgeries, the absolute number of people receiving cataract surgeries is still growing. According to the latest data, the cataract surgery rate in the United States was up to 10,000 in 2018, while in China, the rate was only 2000 to 3000 [24–26].

The main surgical technique in the United States from 1980 to 2004 was PK. However, in the following 10 years, PK dramatically decreased from 95% in 2005 to 42% in 2014 and has been surpassed by various LK techniques (in the form of EK and ALK) in 2011 [4]. Analyzing the results of our survey, we found that from 2014 to 2016 the proportion of PK decreased by 4.71% while LK increased by 4.71%, and then in 2017 and 2018, the proportion of the techniques were relatively stable. In addition, to find the long-term trend of preferred techniques, the authors analyzed a single center study conducted by Shandong Eye Institute, covering 5316 cases of CT from 1996 to 2007 and found that in the twelve years, PK decreased from 76.0% to 61.8% while LK increased from 24.0% to 38.2% [27]. Therefore, it can be seen that the trend of preferred techniques of CT in China and America was similar in the past twenty years, with a decrease in PK and an increase in LK. The change of preferred technique from full thickness PK to LK can reduce the risk of surgery and the risk of immune rejection after surgery. In addition, the application of the new LK technique, such as deep anterior lamellar keratoplasty (DALK) and Descemet's membrane endothelial keratoplasty (DMEK), has made it possible for one or more people to share one donor cornea [28], which can reduce the burden of corneal blindness under the worldwide shortage of donor corneas.

Although in both China and America, the number of LK increased gradually, the respective proportion of ALK and EK and the indications were quite different. According to the statistical report of EBAA in 2014, ALK only accounted for 3.4% of various LK techniques while EK accounted for 96.6%. However, the results of our survey showed that ALK accounted for 83.03% while EK only accounted for 16.97% in China in 2018. The differences may be related to the establishment of eye bank association, the different indications in the two countries and the development of surgical techniques. 1) PK and EK require corneal tissue with high quality. In America, benefiting from the oversupply of donor corneas and the professional management by EBAA, the techniques can be performed with qualified corneas or even pre-cut grafts [29]. However, in China, there is a severe shortage of corneas nationwide [30,31]. No doubt that some eye banks may have corneas more than they need, but due to the lack of a unified management and distribution by a national eye bank association, the spare corneas in one eye bank can not be transferred to another bank for the maximum utilization in time. These corneas are usually dehydrated for a long period of preservation in the eye bank. After dehydration, the corneas can only be used for ALK. 2) In China, the indications for CT are mainly corneal leukoma and bacterial keratitis, which are not good indications for EK, while in America, the main indications for CT, PBK, and Fuchs' dystrophy, are good indications for EK. 3) Due to the lamellar interface created during conventional LK, the postoperative visual acuity may be not as good as that after PK. The technical progresses have made it a reality for new techniques of LK to achieve a comparable visual acuity to PK [32]. In addition, for some diseases that were not typical indications for LK, such as chemical or thermal burns, herpes simplex keratitis scar, and bacterial keratitis scar, DALK is also applicable now, which contributes to the high proportion of LK [33,34].

Additionally, we noticed that the only common indication among the top four indications for CT in China and America was keratoconus, but the preferred techniques for it were quite different. Although there was an increase in ALK for keratoconus in America, PK remained the main surgical method, performed in 6224 patients while ALK only performed in 757 patients in 2014 [4]. In our survey, the proportion of ALK for keratoconus increased from

46.28% in 2014 to 71.37% in 2018 while PK decreased from 53.72% to 28.63%. The common practice of flattening the recipient bed in the process of ALK, which can affect the postoperative visual recovery due to the recipient bed wrinkles, may be a limiting factor for the application of ALK for patients with keratoconus in the past. With proper measures in surgery, the wrinkles can be avoided in the pupil area. Together with the application of DALK, the patients with keratoconus can obtain visual acuity comparable to PK [35]. Benefiting from improvements in techniques, even for cases with acute keratoconus, ALK can be performed before stroma scarring occurs. All these factors have contributed to the high proportion of ALK for keratoconus in China [36].

Due to immune rejection, chronic allograft dysfunction and late graft failure, the average survival time for graft after PK is only about 17 years. However, ALK can significantly lower the risk of chronic graft dysfunction and the graft can survive for about 49 years [37]. As most of the patients with keratoconus are teenagers, an early allograft dysfunction can affect their life. Therefore, ALK (including DALK), should be gradually popularized and applied worldwide for patients with keratoconus.

In our survey, information of CT from 2014 through 2018 in 64 hospitals were included. According to the data collected, the five largest hospital by number of CT surgeries completed 43.68% of the total number of CT surgeries in China and the five largest administrative units by number of CT surgeries performed 54.38% of the total number of CT surgeries in China. Except for No.1 Hospital of Xi'an City, the other 4 hospitals are in eastern China.

The geographical distributions of CT may correlate with the regional economy. The administrative units in eastern China have advantages over the units in central and western China. The rapid development of economy has opened people's mind and increased their health needs. They are more willing to seek medical attention and more active in organ and tissue donation. In addition, the high economy level is the foundation of better transportation and medical level, which can benefit the patients and the cornea procurement.

## Conclusions

In our survey, the data of CT in China from 2014 to 2018 were collected and analyzed to provide information for ophthalmologists at home and abroad to understand the condition of keratoplasty in China and then to conduct further studies. Although we have tried to contact as much hospitals as possible to collect information on CT, it is estimated that about 10% of the CT cases in China were not included. In addition, as this is the first time that a questionnaire has been designed to collect the general information of CT in China, limitations of the questionnaire exist inevitably. In the following five years, the questionnaire will be modified to collect more complete information (including the graft survival time) and use more standardized diagnoses and classification of indications to reflect the condition of CT in China comprehensively.

Overall, in China, the geographical distribution of keratoplasties and the number of keratoplasties performed in each hospital are quite unbalanced. To better serve the patients in central and western China, training for ophthalmologists, development of medical facility and popularization of donation knowledge at these regions are required. PK and ALK are the main surgical techniques for CT in China. Corneal leukoma, bacterial keratitis, and corneal dystrophies are the main indications for keratoplasties. These differences between China and America relate to the national conditions, economic development level, and ethnic characteristics.

## Supporting information

**S1 Table. Questionnaire of keratoplasty in China (Chinese and English version).**
(PDF)

## Acknowledgments

We sincerely thank all the hospitals and doctors who have contributed to the collection of the data.

## Author Contributions

**Conceptualization:** Lixin Xie, Weiyun Shi.

**Data curation:** Hua Gao, Ting Huang, Zhiqiang Pan, Jie Wu, Jianjiang Xu, Jing Hong, Wei Chen, Huping Wu, Qian Kang, Lei Zhu, Lingling Fu, Liqiang Wang, Guigang Li, Zhihong Deng, Hong Zhang, Hui Xu, Qingliang Zhao, Hongshan Liu, Linnong Wang, Baihua Chen, Xiuming Jin, Minghai Huang, Jizhong Yang, Minghong Gao, Wentian Zhou, Hanping Xie, Yao Fu, Feng Wen, Changbo Fu, Shaozhen Zhao, Yanning Yang, Yanjiang Fu, Tao Yao, Chaoqing Wang, Xiaonan Sun, Xiaowei Gao, Maimaitiming Reziwan, Yingping Deng, Jian Li, Limei Liu, Bo Zeng, Lianyun Bao, Hua Wang, Lijun Zhang, Zhiyuan Li, Zhijian Yin, Yuechun Wen, Xiao Zheng, Liqun Du, Zhenping Huang, Xunlun Sheng, Hui Zhang, Lizhong Chen, Xiaoming Yan, Xiaowei Liu, Wenhui Liu, Yuan Liu, Liang Liang, Pengcheng Wu, Lijun Qu, Jinkui Cheng, Hua Zhang, Qige Qi, Yangkyi Tseten, Jianping Ji, Jin Yuan, Ying Jie, Jun Xiang, Yifei Huang, Yuli Yang, Ying Li, Yiyi Hou.

**Formal analysis:** Hua Gao, Lixin Xie, Weiyun Shi.

**Funding acquisition:** Hua Gao, Weiyun Shi.

**Investigation:** Lixin Xie.

**Methodology:** Hua Gao, Weiyun Shi.

**Resources:** Hua Gao, Ting Huang, Zhiqiang Pan, Jie Wu, Jianjiang Xu, Jing Hong, Wei Chen, Huping Wu, Qian Kang, Lei Zhu, Lingling Fu, Liqiang Wang, Guigang Li, Zhihong Deng, Hong Zhang, Hui Xu, Qingliang Zhao, Hongshan Liu, Linnong Wang, Baihua Chen, Xiuming Jin, Minghai Huang, Jizhong Yang, Minghong Gao, Wentian Zhou, Hanping Xie, Yao Fu, Feng Wen, Changbo Fu, Shaozhen Zhao, Yanning Yang, Yanjiang Fu, Tao Yao, Chaoqing Wang, Xiaonan Sun, Xiaowei Gao, Maimaitiming Reziwan, Yingping Deng, Jian Li, Limei Liu, Bo Zeng, Lianyun Bao, Hua Wang, Lijun Zhang, Zhiyuan Li, Zhijian Yin, Yuechun Wen, Xiao Zheng, Liqun Du, Zhenping Huang, Xunlun Sheng, Hui Zhang, Lizhong Chen, Xiaoming Yan, Xiaowei Liu, Wenhui Liu, Yuan Liu, Liang Liang, Pengcheng Wu, Lijun Qu, Jinkui Cheng, Hua Zhang, Qige Qi, Yangkyi Tseten, Jianping Ji, Jin Yuan, Ying Jie, Jun Xiang, Yifei Huang, Yuli Yang, Ying Li, Yiyi Hou.

**Software:** Hua Gao, Tong Liu.

**Supervision:** Lixin Xie, Weiyun Shi.

**Validation:** Hua Gao.

**Visualization:** Hua Gao.

**Writing – original draft:** Hua Gao.

**Writing – review & editing:** Tong Liu, Lixin Xie, Weiyun Shi.

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
