## [Decision Letter · Decision Letter 0]

18 May 2020

PONE-D-20-09119

Keratoplasty in China: a 5-year review from 2014 to 2018

PLOS ONE

Dear Dr. Shi,

Thank you for submitting your manuscript to PLOS ONE. After careful consideration, we feel that it has merit but some points have to be addressed before it can be considered further. Therefore, we invite you to submit a revised version of the manuscript that addresses the points raised during the review process.

We would appreciate receiving your revised manuscript by Jul 02 2020 11:59PM. To enhance the reproducibility of your results, we recommend that if applicable you deposit your laboratory protocols in protocols.io, where a protocol can be assigned its own identifier (DOI) such that it can be cited independently in the future. For instructions see: http://journals.plos.org/plosone/s/submission-guidelines#loc-laboratory-protocols

We look forward to receiving your revised manuscript.

Kind regards,

Yu-Chi Liu, M.D

Academic Editor

PLOS ONE

Journal Requirements:

3. Please modify the title to ensure that it is meeting PLOS’ guidelines (https://journals.plos.org/plosone/s/submission-guidelines#loc-title). In particular, the title should be "specific, descriptive, concise, and comprehensible to readers outside the field" and in this case it is not informative and specific about your study's scope and methodology.

5. Please amend the manuscript submission data (via Edit Submission) to include authors:

Ting Huang2,3, Zhiqiang Pan2,4, Jie Wu2,5, Jianjiang Xu2,6, Jing Hong2,7,

Wei Chen,2,8, Huping Wu9, Qian Kang10, Lei Zhu2,11, Lingling Fu12, Liqiang Wang2,13,

Guigang Li2,14, Zhihong Deng15, Hong Zhang2,16, Hui Xu17, Qingliang Zhao18,

Hongshan Liu19, Linnong Wang2,20, Baihua Chen2,21, Xiuming Jin2,22, Minghai

Huang23, Jizhong Yang24, Minghong Gao2,25, Wentian Zhou26, Hanping Xie2,27, Yao

Fu2,28, Feng Wen29, Changbo Fu30, Shaozhen Zhao2,31, Yanning Yang2,32, Yanjiang

Fu33, Tao Yao34, Chaoqing Wang35, Xiaonan Sun36, Xiaowei Gao2,37, Maimaitiming

Reziwan38, Yingping Deng2,39, Jian Li40, Limei Liu 41, Bo Zeng42, Lianyun Bao43, Hua

Wang 2,44, Lijun Zhang2,45, Zhiyuan Li46, Zhijian Yin47, Yuechun Wen 48, Xiao Zheng

49, Liqun Du 50, Zhenping Huang 51, Xunlun Sheng 52, Hui Zhang2,53, Lizhong Chen 54,

Xiaoming Yan 2,55, Xiaowei Liu 56, Wenhui Liu 57, Yuan Liu 58, Liang Liang 59,

Pengcheng Wu 60, Lijun Qu61, Jinkui Cheng62, Hua Zhang63, Qige Qi64, Yangkyi

Tseten65, Jianping Ji3, Jin Yuan2, 3, Ying Jie4, Jun Xiang6, Yifei Huang2,13, Yuli

Yang2,27, Ying Li2,56, Yiyi Hou1, Tong Liu1, Lixin Xie.

Additional Editor Comments (if provided):

Reviewers' comments:

Reviewer's Responses to Questions

**Comments to the Author**

1. Is the manuscript technically sound, and do the data support the conclusions?

Reviewer #1: Yes

Reviewer #2: Yes

Reviewer #3: Yes

2. Has the statistical analysis been performed appropriately and rigorously? 

Reviewer #1: Yes

Reviewer #2: N/A

Reviewer #3: N/A

3. Have the authors made all data underlying the findings in their manuscript fully available?

Reviewer #1: No

Reviewer #2: Yes

Reviewer #3: Yes

4. Is the manuscript presented in an intelligible fashion and written in standard English?

Reviewer #1: Yes

Reviewer #2: No

Reviewer #3: Yes

5. Review Comments to the Author

Reviewer #1: This is a very extensive study with valuable information. The paper is well written. I have the following comments/suggestions to improve the scientific content.

The information of donor cornea preservation is incomplete. The authors should elaborate this aspect by providing specific information about the type of corneal storage medium and describe what is meant by the dry method.

The survey was based on a questionnaire. More details of the methodology of obtaining the information and the questions asked and collection of the responses can be provided.

Reviewer #2: Thank you for the opportunity to review this manuscript. It has the potential to be an important paper, as this is the first time such a comprehensive survey of keratoplasty in China has been attempted.

The current paper is a good draft, but I think there are multiple clarifications that need to be made before it can be published.

Background

1) many readers outside of China will not be familiar with the geography and the classification of provinces in China, so may I suggest that you explain

- how provinces and cities in China are categorised into "tiers"

- if possible, a map to show where the various participating hospitals are located - you may not be able to place all 64 hospitals on the map, but perhaps you can list the hospitals by geographical location?

Methods

2) Some detail needs to be provided on how corneas in China are obtained.

- I understand that use of foreign or imported tissue is highly restricted?

- the use of bio-engineered tissue is very interested, could you elaborate on how these tissues are engineered?

- proportion of fresh, versus frozen, versus bio-engineered tissue would give readers some context

3) a copy of the questionnaire used will be important, and included as an annex. An English translation would be appropriate for most readers.

4) Was there also information collected on graft survival, at least in the first year?

5) Information on the number of surgeries done - lines 276 to 296 - is best summarised in a table rather than written in the text. Similarly with the gender and age distribution.

6) Indications for surgery

- unfortunately, cornea leukoma is a rather rather term, unless there is a specific-definition used by the Chinese Cornea Society? Usually there is an underlying cause - congenital, trauma, infection. Would the authors be able to give further details?

- similarly, cornea perforation is very common, but it would be better to further tell readers if this is due to trauma or infection, ot other causes

- I am surprised bacteria keratitis is such a common indication, but fungal keratitis is not, even though fungal infections are as high as bacterial keratitis in China. Is this practice pattern (meaning that surgery is not done for fungal keratitis), or is it because it is classified under other indications - eg perforations?

7) ALK indications

- curious to see small numbers of corneal perforation - are these patch grafts?

- similarly I see ALK for pseuophakic bullous keratopathy

8) Discussion

- Could the authors comment how the availability of tissue also affects practice patterns?

for example, there could be much more ALK done in China vs the USA because of the lack of healthy fresh tissue for PK and EK

- could authors also comment on what more can be done to improve the CT situation - what this paper shows is that much more CT needs to be done to have any hope of helping the 3 million in China with cornea blindness - is bioengineered cornea a good option? what about training and facilities - 64 hospitals perform transplants, but many are done in smaller units and do much fewer transplants than the top 5 hospitals

Reviewer #3: This paper aims to analyze the trend of corneal transplants in China, focusing in particular on the diffusion of lamellar techniques and surgical indications and to compare the results with the trend in United States (US).

I found the manuscript interesting and original, since few data are available about corneal transplant in China.

I suggests the authors to include the survey as supplemental material.

Moreover, I suggest to include in the discussion the results of similar studies conducted in other countries, in addiction to the US situation.

The sentence at page 21 ,lines 377-328, should be corrected, since only EK surpassed the number of PK in US since 2011 (ALK is still inferior to PK).

6. PLOS authors have the option to publish the peer review history of their article (what does this mean?). If published, this will include your full peer review and any attached files.

Reviewer #1: Yes: Dr Radhika Tandon, MD, FRCOphth, FRCSEd; Professor of Ophthalmology, Co-Chairperson National Eye Bank, Dr Rajendra Prasad Centre for Ophthalmic Sciences, All India Institute of Medical Sciences, New Delhi, India. Also immediate past president Eye Bank Association of India (EBAI) and immediate past Vice President Association of Eye Banks of Asia (AEBA).

Reviewer #2: No

Reviewer #3: No

---

## [Author Response · Author response to Decision Letter 0]

1 Jul 2020

Dear editors and reviewers,

We are grateful to you for your time and constructive comments on our manuscript entitled "Survey report on keratoplasty in China: a 5-year review from 2014 to 2018". The comments and suggestions are very valuable and useful for improving the manuscript. We have carefully implemented the comments and suggestions and wish to submit this revised version for further consideration in the journal. Below, we also provide point-by-point responses explaining how we have addressed each of the comments. 

Looking forward to hearing from you soon.

Thank you and with kindest regards.

Yours Sincerely,

Weiyun Shi

On behalf of co-authors

Journal Requirements:

Answer： We have modified the manuscript according to the templates. 

Answer： We have included a copy of the questionnaire in both Chinese and English as supporting information.

3. Please modify the title to ensure that it is meeting PLOS’ guidelines (https://journals.plos.org/plosone/s/submission-guidelines#loc-title). In particular, the title should be "specific, descriptive, concise, and comprehensible to readers outside the field" and in this case it is not informative and specific about your study's scope and methodology.

Answer: We have modified the title to “Survey report on keratoplasty in China: a 5-year review from 2014 to 2018” according to the guidelines. If the title should be further changed, would you please give us some suggestion? Thank you very much.

Answer: The ORCID ID of the corresponding author has been added. Thank you. 

5. Please amend the manuscript submission data (via Edit Submission) to include authors:

Ting Huang2,3, Zhiqiang Pan2,4, Jie Wu2,5, Jianjiang Xu2,6, Jing Hong2,7,

Wei Chen,2,8, Huping Wu9, Qian Kang10, Lei Zhu2,11, Lingling Fu12, Liqiang Wang2,13,

Guigang Li2,14, Zhihong Deng15, Hong Zhang2,16, Hui Xu17, Qingliang Zhao18,

Hongshan Liu19, Linnong Wang2,20, Baihua Chen2,21, Xiuming Jin2,22, Minghai

Huang23, Jizhong Yang24, Minghong Gao2,25, Wentian Zhou26, Hanping Xie2,27, Yao

Fu2,28, Feng Wen29, Changbo Fu30, Shaozhen Zhao2,31, Yanning Yang2,32, Yanjiang

Fu33, Tao Yao34, Chaoqing Wang35, Xiaonan Sun36, Xiaowei Gao2,37, Maimaitiming

Reziwan38, Yingping Deng2,39, Jian Li40, Limei Liu 41, Bo Zeng42, Lianyun Bao43, Hua

Wang 2,44, Lijun Zhang2,45, Zhiyuan Li46, Zhijian Yin47, Yuechun Wen 48, Xiao Zheng

49, Liqun Du 50, Zhenping Huang 51, Xunlun Sheng 52, Hui Zhang2,53, Lizhong Chen 54,

Xiaoming Yan 2,55, Xiaowei Liu 56, Wenhui Liu 57, Yuan Liu 58, Liang Liang 59,

Pengcheng Wu 60, Lijun Qu61, Jinkui Cheng62, Hua Zhang63, Qige Qi64, Yangkyi

Tseten65, Jianping Ji3, Jin Yuan2, 3, Ying Jie4, Jun Xiang6, Yifei Huang2,13, Yuli

Yang2,27, Ying Li2,56, Yiyi Hou1, Tong Liu1, Lixin Xie.

Answer: Thank you very much. Information of all the authors has been added to the submission system. 

Answer: Thank you very much. We have included the caption for the supporting information at the end of the manuscript and updated the in-text citation. 

Reviewers' comments:

Comments to the Author

Reviewer #1: This is a very extensive study with valuable information. The paper is well written. I have the following comments/suggestions to improve the scientific content.

The information of donor cornea preservation is incomplete. The authors should elaborate this aspect by providing specific information about the type of corneal storage medium and describe what is meant by the dry method.

Answer: Thank you very much for raising this point. Limited by the requirements on article length, we have not given comprehensive information of donor cornea preservation in the manuscript that we submitted. In fact, there is much information that should be reported, so after careful discussion, in this revised manuscript, we decided to delete all the information about donor cornea preservation and in the near future, we will write an article reporting the survey results on cornea preservation and compare them with those in America. 

The survey was based on a questionnaire. More details of the methodology of obtaining the information and the questions asked and collection of the responses can be provided.

Answer: Thank you very much for your suggestion. In the revised manuscript, we have added the questionnaire as supporting information. 

Reviewer #2: Thank you for the opportunity to review this manuscript. It has the potential to be an important paper, as this is the first time such a comprehensive survey of keratoplasty in China has been attempted.

The current paper is a good draft, but I think there are multiple clarifications that need to be made before it can be published.

Background

1) many readers outside of China will not be familiar with the geography and the classification of provinces in China, so may I suggest that you explain

- how provinces and cities in China are categorised into "tiers"

- if possible, a map to show where the various participating hospitals are located - you may not be able to place all 64 hospitals on the map, but perhaps you can list the hospitals by geographical location?

Answer: Thank you very much for your suggestion. We agree with you on your opinion that many foreigners are not familiar with the classification of provinces in China, so in the manuscript, to avoid confusing the readers, we use “provincial-level administrative units” instead of terms like “province” and “municipality directly under the central government”. 

Methods

2) Some detail needs to be provided on how corneas in China are obtained.

- I understand that use of foreign or imported tissue is highly restricted?

- the use of bio-engineered tissue is very interested, could you elaborate on how these tissues are engineered?

- proportion of fresh, versus frozen, versus bio-engineered tissue would give readers some context

Answer: Thank you very much for raising this point. As you said, the use of foreign and imported tissue is highly restricted. In China, the procurement, management and use of cornea tissue are more standardized in recent years. However, there is still a huge shortage of corneas in China, so ophthalmologists are making efforts to look for alternatives. Prof. Weiyun Shi and his team have made breakthrough in the field of bio-engineered cornea and published articles about decellularized porcine cornea (Shi, W, Zhou, Q, Gao, H, Li, S, Dong, M, & Wang, T, et al. (2019). Protectively decellularized porcine cornea versus human donor cornea for lamellar transplantation. Advanced Functional Materials, 29(37), 1902491.1-1902491.12. Dong, Muchen & Zhao, Long & Wang, Fuyan & Hu, Xiaoli & Li, Hua & Liu, Ting & Zhou, Qingjun & Shi, Weiyun. (2019). Rapid porcine corneal decellularization through the use of sodium N-lauroyl glutamate and supernuclease. Journal of Tissue Engineering. 10. 204173141987587. 10.1177/2041731419875876.). In our study, the main preservation methods of donor corneas were mid-term preservation (29 252, 79.89%), preservation by dehydration (5 864, 16.01%) and preservation in moist chamber (1 500, 4.10%). We found that there is much information that can be reported based on the results of the survey, so we decided to delete all the information about cornea procurement and preservation in this article and write an article thoroughly reporting the information. 

3) a copy of the questionnaire used will be important, and included as an annex. An English translation would be appropriate for most readers.

Answer: Thank you very much for your suggestion. We have added a copy of questionnaire as supporting information.

4) Was there also information collected on graft survival, at least in the first year?

Answer: Thank you very much for raising this point. To be honest, we have not collected information on graft survival. In the future, when we do this survey, we will improve the questionnaire and collect information on CT. 

5) Information on the number of surgeries done - lines 276 to 296 - is best summarised in a table rather than written in the text. Similarly with the gender and age distribution.

Answer: Thank you very much for your suggestion. We have expressed the information in a table. 

6) Indications for surgery

- unfortunately, cornea leukoma is a rather rather term, unless there is a specific-definition used by the Chinese Cornea Society? Usually there is an underlying cause - congenital, trauma, infection. Would the authors be able to give further details?

Answer: Thank you very much for raising this point. According to the survey results collected from the hospitals, we can not give detailed information on the underlying causes. We have called the hospital and they said that for some patients, the underlying causes were hard to diagnosed, so cornea leukoma was the indication for their surgeries. The cases whose causes of cornea leukoma were diagnosed have been included in the corresponding indications. 

- similarly, cornea perforation is very common, but it would be better to further tell readers if this is due to trauma or infection, ot other causes

Answer: Thank you very much for raising this point. According to the survey results collected from the hospitals, we can not give detailed information on the underlying causes. We have called the hospital and they said that for some patients, the underlying causes were hard to diagnosed, so cornea perforation was the indication for their surgeries. The cases whose causes of cornea perforation were diagnosed have been included in the corresponding indications.

- I am surprised bacteria keratitis is such a common indication, but fungal keratitis is not, even though fungal infections are as high as bacterial keratitis in China. Is this practice pattern (meaning that surgery is not done for fungal keratitis), or is it because it is classified under other indications - eg perforations?

Answer: Thank you very much for your question. In the survey, fungal keratitis was not classified under other indications. 

7) ALK indications

- curious to see small numbers of corneal perforation - are these patch grafts?

- similarly I see ALK for pseuophakic bullous keratopathy

Answer: Thank you very much for raising this point. The main reason for ALK performed for patients with corneal perforation and pseuophakic bullous keratopathy was the shortage of fresh cornea tissue in China, so ALK was used as an alternative to save the patients’ eyes. Besides, for patients with pseuophakic bullous keratopathy, PK and EK are the preferred methods, but without qualified corneas, ALK can be performed to alleviate their pain.

8) Discussion

- Could the authors comment how the availability of tissue also affects practice patterns?

for example, there could be much more ALK done in China vs the USA because of the lack of healthy fresh tissue for PK and EK 

Answer: Thank you very much for raising this point. The cornea types that are available do influence the surgery types the ophthalmologists selected. Benefiting from the questions and suggestions from the reviewers, we realized that we need to reporting more valuable information on cornea procurement and preservation, which influences the surgery patterns selected. We should not waste the hard-won information from the survey, so in addition to this article, we decided to write another article reporting this aspect comprehensively. 

- could authors also comment on what more can be done to improve the CT situation - what this paper shows is that much more CT needs to be done to have any hope of helping the 3 million in China with cornea blindness - is bioengineered cornea a good option? what about training and facilities - 64 hospitals perform transplants, but many are done in smaller units and do much fewer transplants than the top 5 hospitals

Answer: Thank you very much for raising these points. Measures should be taken to improve the CT situation in China. For example, the society can pay more attention to the publicity of cornea donation and the government and professional organizations can carry out more training. A course on corneal transplantation led by Prof. Weiyun Shi has training more than 200 ophthalmologists. 

Bio-engineered cornea is a good option, which can ease the situation of cornea shortage in China. Bio-engineered corneas do not need to be stored in eye banks, so it is more convenient to use. 

To narrow the gap between the hospitals, we think that developing economy is of great importance. 

Reviewer #3: This paper aims to analyze the trend of corneal transplants in China, focusing in particular on the diffusion of lamellar techniques and surgical indications and to compare the results with the trend in United States (US).

I found the manuscript interesting and original, since few data are available about corneal transplant in China.

I suggests the authors to include the survey as supplemental material.

Moreover, I suggest to include in the discussion the results of similar studies conducted in other countries, in addiction to the US situation.

The sentence at page 21 ,lines 377-328, should be corrected, since only EK surpassed the number of PK in US since 2011 (ALK is still inferior to PK).

Answer: Thank you very much for your careful review. We have added the questionnaire as supporting information. 

Comparing the results in China with the results in other countries can better reflect the differences and provide more information for readers, but this survey was designed referring to the EBAA report, so we only compared the results with those in America. At the beginning of the study, we have compared reports and articles of different countries in this field. EBAA was the one who reported comprehensively and annually, so we decided to compare the situation in China with the situation in America. 

In the sentence at page 21, lines 377-378, “various LK” means ALK and EK. Thank you for pointing out. To express the meaning more clearly, we have modified the expression in the sentence.

---

## [Decision Letter · Decision Letter 1]

3 Aug 2020

PONE-D-20-09119R1

Survey report on keratoplasty in China: a 5-year review from 2014 to 2018

PLOS ONE

Dear Dr. Shi,

Thank you for submitting your manuscript to PLOS ONE. The reviewers have raised some comments and we would like to invite you to submit a revised version of the manuscript that addresses the points raised during the review process.

We look forward to receiving your revised manuscript.

Kind regards,

Yu-Chi Liu, M.D

Academic Editor

PLOS ONE

Reviewers' comments:

Reviewer's Responses to Questions

**Comments to the Author**

1. If the authors have adequately addressed your comments raised in a previous round of review and you feel that this manuscript is now acceptable for publication, you may indicate that here to bypass the “Comments to the Author” section, enter your conflict of interest statement in the “Confidential to Editor” section, and submit your "Accept" recommendation.

Reviewer #1: All comments have been addressed

Reviewer #2: (No Response)

Reviewer #3: All comments have been addressed

2. Is the manuscript technically sound, and do the data support the conclusions?

Reviewer #1: (No Response)

Reviewer #2: Yes

Reviewer #3: Yes

3. Has the statistical analysis been performed appropriately and rigorously? 

Reviewer #1: (No Response)

Reviewer #2: Yes

Reviewer #3: N/A

4. Have the authors made all data underlying the findings in their manuscript fully available?

Reviewer #1: (No Response)

Reviewer #2: Yes

Reviewer #3: Yes

5. Is the manuscript presented in an intelligible fashion and written in standard English?

Reviewer #1: (No Response)

Reviewer #2: No

Reviewer #3: Yes

6. Review Comments to the Author

Reviewer #1: The authors have addressed the concerns raised. Minor editorial corrections and language changes are required. For example one-center can be changed to single center. Line 468-471 the sentence "Overall, in China, the geographical ...weak in CT" in the conclusion needs to be re-written as it is not clear.

Reviewer #2: Thank you once again for the opportunity to review this paper, as I was Reviewer 2 for the first submission.

I appreciate that the authors have made some of the amendments as I have suggested, including summarising the data in tables.

Other information that I felt would have made this a more complete paper appears to not be available, based on the methods and on how the questionnaire was designed. I think it is therefore important to include a section of the paper to address the limitations of the study - namely that this was a retrospective study where information is incomplete, diagnoses and classification of indications are not standardised (eg the meaning of cornea leukoma), and that outcome data in terms of graft survival was not obtained.

Reviewer #3: (No Response)

7. PLOS authors have the option to publish the peer review history of their article (what does this mean?). If published, this will include your full peer review and any attached files.

Reviewer #1: No

Reviewer #2: No

Reviewer #3: No

---

## [Author Response · Author response to Decision Letter 1]

19 Aug 2020

Dear Dr. Liu and reviewers,

We would like to give our great thanks to you and all the reviewer, for your valuable comments on our manuscript entitled "Survey report on keratoplasty in China: a 5-year review from 2014 to 2018". Thank you for your suggestions and efforts to make the article better. We have carefully implemented the comments and suggestions and wish to submit this revised version for further consideration in the journal. Below, we provide point-by-point responses explaining how we have addressed each of the comments.

Looking forward to hearing from you soon.

Thank you and with kindest regards.

Yours Sincerely,

Weiyun Shi

On behalf of co-authors

1. Reviewer #1: The authors have addressed the concerns raised. Minor editorial corrections and language changes are required. For example, one-center can be changed to single center. Line 468-471 the sentence "Overall, in China, the geographical ...weak in CT" in the conclusion needs to be re-written as it is not clear.

Answer: Thank you very much for your time and suggestions. We have read the article carefully and thoroughly to avoid typographical and grammatical errors. “One-center” has been changed to “single center”. The sentence has been rewritten. 

2. Reviewer #2: Thank you once again for the opportunity to review this paper, as I was Reviewer 2 for the first submission. I appreciate that the authors have made some of the amendments as I have suggested, including summarizing the data in tables. Other information that I felt would have made this a more complete paper appears to not be available, based on the methods and on how the questionnaire was designed. I think it is therefore important to include a section of the paper to address the limitations of the study - namely that this was a retrospective study where information is incomplete, diagnoses and classification of indications are not standardized (eg the meaning of cornea leukoma), and that outcome data in terms of graft survival was not obtained.

Answer: Thank you very much for your valuable suggestions on the previous manuscripts. We benefit a lot from your comments and we have modified the article accordingly. No doubt there are limitations in the questionnaire and they are hard to be corrected for the survey from 2014 to 2018. Therefore, we adopted your advice and added sentences at the end of the manuscript to address the limitations of the study. We appreciate your efforts to make the article and the questionnaire better.

---

## [Decision Letter · Decision Letter 2]

16 Sep 2020

Survey report on keratoplasty in China: a 5-year review from 2014 to 2018

PONE-D-20-09119R2

Dear Dr. Shi,

We’re pleased to inform you that your manuscript has been judged scientifically suitable for publication and will be formally accepted for publication once it meets all outstanding technical requirements.

Kind regards,

Yu-Chi Liu, M.D

Academic Editor

PLOS ONE

Additional Editor Comments (optional):

Reviewers' comments:

Reviewer's Responses to Questions

**Comments to the Author**

1. If the authors have adequately addressed your comments raised in a previous round of review and you feel that this manuscript is now acceptable for publication, you may indicate that here to bypass the “Comments to the Author” section, enter your conflict of interest statement in the “Confidential to Editor” section, and submit your "Accept" recommendation.

Reviewer #2: All comments have been addressed

2. Is the manuscript technically sound, and do the data support the conclusions?

Reviewer #2: Yes

3. Has the statistical analysis been performed appropriately and rigorously? 

Reviewer #2: Yes

4. Have the authors made all data underlying the findings in their manuscript fully available?

Reviewer #2: Yes

5. Is the manuscript presented in an intelligible fashion and written in standard English?

Reviewer #2: Yes

6. Review Comments to the Author

Reviewer #2: (No Response)

7. PLOS authors have the option to publish the peer review history of their article (what does this mean?). If published, this will include your full peer review and any attached files.

Reviewer #2: No

---

## [Editor Report · Acceptance letter]

5 Oct 2020

PONE-D-20-09119R2 

Survey report on keratoplasty in China: a 5-year review from 2014 to 2018 

Dear Dr. Shi:

I'm pleased to inform you that your manuscript has been deemed suitable for publication in PLOS ONE. Congratulations! Your manuscript is now with our production department. 

Kind regards, 

on behalf of

Dr. Yu-Chi Liu 

Academic Editor

PLOS ONE